DATA RELEASE

# Collection of entomological, demographic, water and sanitation, and climatic data of interest for arbovirus surveillance in Praia, Cabo Verde

Lara Ferrero Gómez[1,*], Keily Lucienne Fonseca Silva[1,2],
Bruno dos Santos Pina[3], Patrick Silva[4], Ulisses António Lima da Cruz[5],
José Moniz Lopes Fernandes[4] and Hélio Daniel Ribeiro Rocha[1,6]

1 Grupo de Investigação em Doenças Tropicais (GIDTPiaget), Unidade das Ciências da Vida, da Natureza e do Ambiente (UNCVA), Universidade Jean Piaget de Cabo Verde, Praia, Cabo Verde, Palmarejo Grande, Praia, 7600, Cabo Verde
2 Departamento de Entomologia, Instituto Aggeu Magalhães/Fundação Oswaldo Cruz (FIOCRUZ-PE), Av. Professor Moraes Rego s/n, Cidade Universitária, Recife, PE 50670-420, Brasil
3 Instituto de Higiene e Medicina Tropical (IHMT), Universidade NOVA de Lisboa, Rua da Junqueira, 100, 1349-008 Lisboa, Portugal
4 Facultade de Ciência e Tecnologia (FCT), Universidade de Cabo Verde (UniCV), Campus do Palmarejo Grande, Praia, 7943-010, Cabo Verde
5 Instituto Nacional de Estadística (INE), Rua da Caixa Económica nº 18, Fazenda, Praia, 7600, Cabo Verde
6 Instituto Nacional de Saúde Pública (INSP), Largo do Desastre da Assistência, Chã de Areia, Praia, 7600, Cabo Verde

**Submitted:** 17 September 2025

\* Corresponding author. E-mail: lfg@cv.unipiaget.org; lara.ferrero.gomez@gmail.com

Preprint submitted at https://doi.org/10.5281/zenodo.17250949

Included in the series: *Vectors of human disease* (https://doi.org/10.46471/GIGABYTE_SERIES_0002)

## ABSTRACT

Vector-borne diseases, primarily those transmitted by mosquitoes, are a serious public health problem. Some, such as dengue, put half of the world's population at risk. Combating these diseases requires multifaceted strategies, with vector surveillance and control playing key roles. Robust and predictive surveillance systems for vector-borne diseases, based on risk stratification, enable the implementation of appropriate interventions across time and space. Here, we present a collection of entomological, demographic, water and sanitation, and climatic data from Praia (Cabo Verde), a hotspot for mosquito-borne diseases. These data were collected from June to November 2022, at 40 sentinel points scattered across the urban area of Praia. They constitute a valuable source of information for developing predictive scenarios of arbovirus outbreak risk using statistical models applied to spatial and non-spatial indicators. These data demonstrate the utility of GBIF in transforming large volumes of occurrence data into valuable information for arbovirus surveillance and vector control.

**Subjects** Ecology, Biodiversity, Taxonomy

## DATA DESCRIPTION

### Background and context

Vector-borne diseases pose a health risk to more than 80% of the world's population and represent 17% of the global burden of communicable diseases [1].

For sub-Saharan Africa, diseases transmitted by mosquito vectors, such as yellow fever, dengue, chikungunya, Zika, Rift Valley fever, West Nile virus fever, and malaria, are a

challenge, especially the latter, which is responsible for a disproportionate number of cases and deaths (94% of the 600,000 deaths in 2023) [2]. At the same time, arbovirus epidemics have also grown in the African region. Increased urbanization and climate change favor the proliferation of mosquito vectors, increasing the risk of outbreaks and the need for improved vector surveillance and control [3]. Major epidemics of dengue, Zika, and chikungunya have recently occurred in countries across the region, with more than 27,000 cases of *Aedes*-borne diseases documented since 2007 [4]. The World Health Organization launched the Global Arbovirus Initiative in 2022, a six-pillar international program to combat arbovirus infections. This initiative aims to reduce local risks, stimulate innovation, and strengthen collaborations [5].

Effective vector management requires multidisciplinary data that integrates entomological information with epidemiological, environmental, climatic, and social information to design and implement integrated vector management strategies, such as vector and disease surveillance. An ideal integrated arbovirus surveillance system includes both vector and disease monitoring systems, while also connecting environmental, climatic, and social change monitoring [6]. For surveillance to be effective, it is important to contextualize the physical space and time in which it will be conducted, as well as to define appropriate, measurable indicators to provide evidence that allows predicting the risk of outbreaks of arboviruses, such as dengue and Zika.

Cabo Verde, an archipelagic country in West Africa, lies at the crossroads of three continents (the Americas, Africa, and Europe). Due to its geographic location, climate change, and the effects of intense human and commercial trafficking, Cabo Verde is extremely vulnerable to the emergence and resurgence of vector-borne diseases [7]. Mosquito-borne diseases have been part of its history, with epidemics of malaria and yellow fever since the 16th century and, more recently, dengue fever and Zika [8–12]. The main vectors of arboviruses and malaria are the *Aedes aegypti* and *Anopheles arabiensis* mosquito species, respectively [13–16]. Furthermore, populations of mosquitoes from the *Culex pipiens* s.l, a potential vector for other arboviruses such as West Nile fever and Rift Valley fever, are found on all the islands [17]. Meanwhile, the country's capital, Praia, with a quarter of the total population, has the most significant social inequalities and has also been the most affected by vector-borne diseases [18].

In this pilot study, entomological, climatic, environmental, demographic, and anthropic indicators were collected weekly for six months (from June 7 to November 15, 2022) at forty sentinel points in the capital of the country, to obtain reliable data that can be used to predict spatiotemporally the risk of dengue, Zika, or other emerging arboviruses.

## METHODS

### Data collection

Data were collected in Praia, the capital of Cabo Verde, located on the island of Santiago, which has more than a quarter of the country's population (151,155 inhabitants), according to [19] (2022) (Figure 1). The geolocation of the ovitraps (OVT) was used as a reference point for the collection of all indicators at each sentinel point.

To ensure the representativeness of the entire city in the collected data series, 40 sentinel points were defined, organized into five zones (A, B, C, D, and E) that comprise the five health districts connecting the five health centers with different neighborhoods in the city (Table 1). Sentinel sites were selected considering several factors: locations with a high



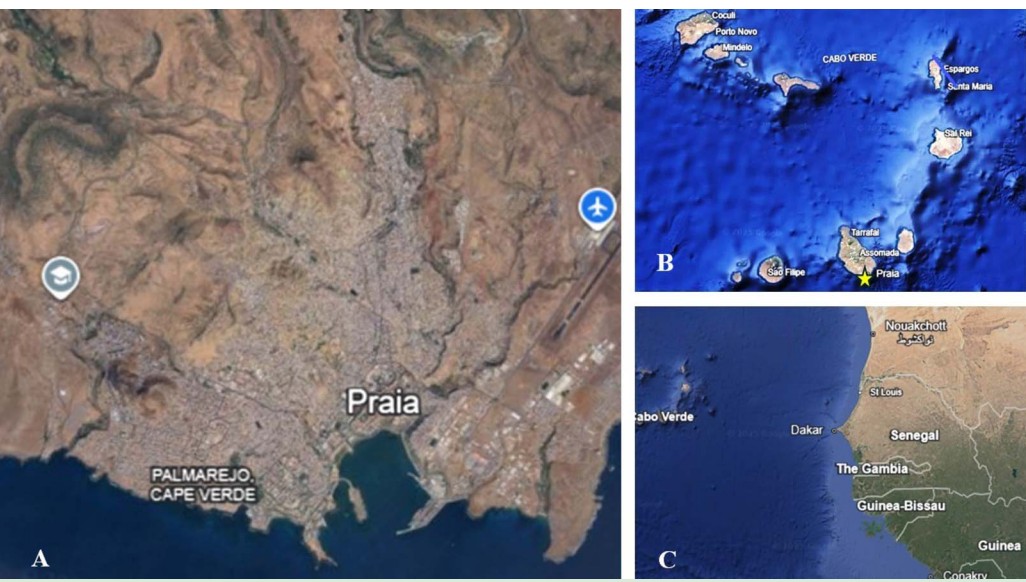

**Figure 1.** (A) Praia, (B) Cape Verde, (C) West Africa region in front of Cabo Verde.

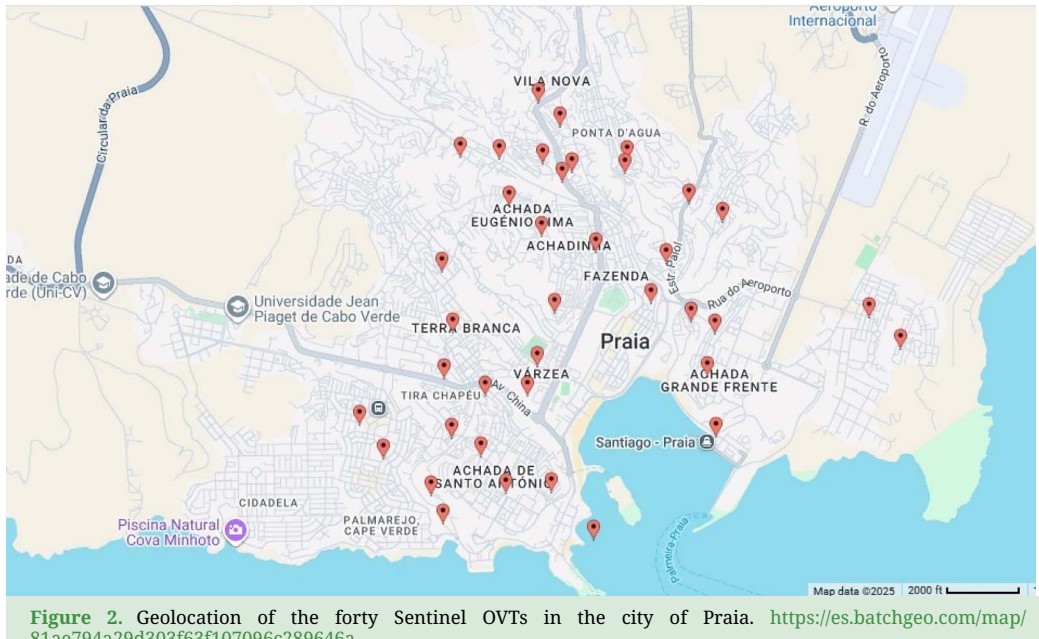

**Figure 2.** Geolocation of the forty Sentinel OVTs in the city of Praia. https://es.batchgeo.com/map/81ae794a29d303f63f107096c289646a

density of vulnerable population, a known history of locations with abundant *Ae. aegypti*, locations with the presence of vegetation, animals, water reservoirs, and protection from the wind.

The forty sentinel points were georeferenced, based on the location of the forty OVTs used in this study as an entomological tool for *Ae. aegypti* egg collection (Figure 2).

Data were collected using the application "ODK Collect" for Android (RRID:SCR_027538) [20] for field data collection and direct transfer to the automatically



**Table 1.** Location of OVTs, grouped by area of coverage of the Health Centers in Praia.

| Zone | Locality | Ref_OVT | Latitude | Longitude |
|---|---|---|---|---|
| A | Marrocos | OVT-1 | 14.91905 | −23.4856 |
| | A. G. Trás Health Center | OVT-2 | 14.92145 | −23.4881 |
| | EB School No. 3 - A.G. Frente | OVT-3 | 14.91702 | −23.5009 |
| | Source of A.G. Frente | OVT-4 | 14.92025 | −23.5003 |
| | Lem Ferreira | OVT-5 | 14.91233 | −23.5002 |
| | Praia Port Administration | OVT-6 | 14.92117 | −23.5022 |
| | Achada Mato School | OVT-7 | 14.93020 | −23.5023 |
| | Castelão Nursing Home | OVT-8 | 14.92882 | −23.4996 |
| B | Ponta D'Agua Health Center | OVT-9 | 14.93255 | −23.5074 |
| | EBI School No. 28 - Ponta D'Agua | OVT-10 | 14.93347 | −23.5172 |
| | Constantino Semedo Secondary School | OVT-11 | 14.94933 | −23.5147 |
| | EBI Julia Costa School | OVT-12 | 14.95578 | −23.5171 |
| | Safende | OVT-13 | 14.93792 | −23.5143 |
| | Safende School | OVT-14 | 14.93612 | −23.5125 |
| | Vila Nova Elementary School | OVT-15 | 14.93263 | −23.5016 |
| | EB School No. 6 - Lém Cachorro | OVT-16 | 14.92567 | −23.5042 |
| C | Manoel Lopes Secondary School | OVT-17 | 14.93187 | −23.5124 |
| | Calabaceira | OVT-18 | 14.93328 | −23.5139 |
| | Pensamento | OVT-19 | 14.93358 | −23.5174 |
| | Kindergarten Pensamento | OVT-20 | 14.93377 | −23.5205 |
| | Eugenio Lima | OVT-21 | 14.93002 | −23.5166 |
| | EBI Eugenio Lima School | OVT-22 | 14.92773 | −23.5143 |
| | Quintino Lopes Elementary School | OVT-23 | 14.92183 | −23.5129 |
| | Capelinha School - Fazenda | OVT-24 | 14.92647 | −23.5097 |
| D | Amor de Deus School, Cabo Verde | OVT-25 | 14.91328 | −23.5284 |
| | EBC Terra Branca School | OVT-26 | 14.91072 | −23.5265 |
| | Tira Chapeau Health Center | OVT-27 | 14.92028 | −23.5211 |
| | Tira Chapeau | OVT-28 | 14.92495 | −23.5219 |
| | Abílio Duarte Secondary School | OVT-29 | 14.91685 | −23.5217 |
| | EBI School 13 de Janeiro | OVT-30 | 14.91230 | −23.5212 |
| | Cemetery | OVT-31 | 14.91777 | −23.5144 |
| | Cónego Jacinto Secondary School | OVT-32 | 14.91553 | −23.5151 |
| E | Atanasio's Garden - Fonton | OVT-33 | 14.90788 | −23.5228 |
| | Palmarejo WWTP | OVT-34 | 14.90570 | −23.5218 |
| | Capelinha School - Tira Chapéu | OVT-35 | 14.91088 | −23.5188 |
| | Achada Santo António Health Center | OVT-36 | 14.91551 | −23.5184 |
| | Pedro Gomes Secondary School | OVT-37 | 14.90807 | −23.5168 |
| | Brasil | OVT-38 | 14.90815 | −23.5133 |
| | Sucupira | OVT-39 | 14.90448 | −23.5099 |
| | Domingos Ramos Secondary School | OVT-40 | 14.92258 | −23.5054 |

**Note**: Zone **A**: A.G. Trás Health Center; Zone **B**: Ponta D'Água Health Center; Zone **C**: Achadinha Health Center; Zone **D**: Tira Chapeu Health Center; Zone **E**: Achada Santo António Health Center.

generated matrix on a computer (Kobo Toolbox) [21]. Data were then transferred directly to the matrix automatically generated on the computer using the Kobo Toolbox.

Fifteen data categories were collected, positionally linked to each georeferenced OVT, and classified into four groups of indicators: entomological, demographic, climatic, and anthropogenic. Some data categories correspond to fixed or static information, while others correspond to dynamic or variable information throughout the study period. Within the dynamic data categories, the data were recorded weekly, except for entomological data from the collection of adult specimens, which were biweekly (Table 2).

## Mosquito collection and identification

For the collection of adult mosquitoes, BG-Sentinel traps were used [22], with BG-Lure scent baits. These traps were kept in the field for 24 h, providing datasets from the 40 sentinel sites for biweekly periods, covering 20 sites per week.

For the collection of *Ae. aegypti* eggs, OVTs adapted from [23] (1965 model) were used, using polyethylene terephthalate bottles as containers and wooden pallets as oviposition

**Table 2.** Data collection characterization by category, group of indicators, and type and frequency of records.

| Data categories | Group of indicators | Data record type | Frequency of data recording |
|---|---|---|---|
| Number of *Ae. aegypti* females | Entomologic | Dynamic | Biweekly |
| Number of *Ae. aegypti* females | Entomologic | Dynamic | Biweekly |
| Number of *Cx. pipiens* s.l. females | Entomologic | Dynamic | Biweekly |
| Number of *Cx. pipiens* s.l. males | Entomologic | Dynamic | Biweekly |
| Number of *Ae. aegypti* eggs | Entomologic | Dynamic | Weekly |
| Average air temperature | Climatic | Dynamic | Weekly |
| Air temperature (°C) | Climatic | Dynamic | Weekly |
| Air humidity (%) | Climatic | Dynamic | Weekly |
| Wind speed (Km/h) | Climatic | Dynamic | Weekly |
| Precipitation (mm$^3$) | Climatic | Dynamic | Weekly |
| Building number (100 m radio) | Demographic | Static | Once |
| Population density in Sentinel OVT | Demographic | Static | Once |
| Population density (100 m radio) | Demographic | Static | Once |
| Water storage type | Anthropogenic | Static | Once |
| Sanitation type | Anthropogenic | Static | Once |

substrates. Forty OVTs filled with tap water were placed weekly outdoors in the 40 sentinel sites, and the pallets were collected after five days. In this study, the OVTs were exclusively placed outdoors, considering the history (from other previous works made by the Tropical Diseases Research Group – GIDTPiaget [24]) of areas and local hotspots of *Ae. aegypti*. Hence, we prioritized high-density localities, including twenty-one educational centers, four health centers, and a nursing home.

Once the field material was collected, it was packaged in thermal bags and taken to the entomology laboratory of the GIDTPiaget for conditioning and identification.

Adult mosquito samples were placed at −20 °C for 20 min to ensure their killing. The specimens were then manually counted, females were separated from males, and morphological differentiation was performed down to the species *(Ae. aegypti)* and species complex (*Cx. pipiens* s.l. and *Anopheles gambiae* s.l.) level using a Motic (SMZ-168) stereoscope and the taxonomic key for Cabo Verde mosquitoes by [8] (1980).

In the laboratory, the pallets collected in the field were dried in a vertical position, without touching each other and uncovered, for 24 h. Next, egg counting was conducted by transects of pallet areas under the stereoscope mentioned above.

## Statistical analysis

A descriptive analysis was performed using Excel 2021 and XLSTAT 2025, a statistical software for Excel, to present the quantitative results of the entomological data series collected in the field. This data included the distribution and abundance of adult mosquito species and eggs of *Ae. Aegypti* (NCBI:txid7159). The presentation of the measure was done through tables. Two different indices were used to estimate the vector population density and its distribution: the positive OVT index (OI), which is the proportion of OVTs positive for the presence of *Aedes* eggs, and the density eggs index (EDI), which is the average number of eggs per positive OVT used to estimate the vector population infestation [25]. A Pearson correlation coefficient was applied, with statistical significance set at $p \leq 0.05$, to analyze the relationship between the observed profiles, throughout the study period, for the data series and entomological indices.

## DATA VALIDATION AND QUALITY CONTROL

Initially, all data collected in the field were reviewed and validated in real-time with the open-source platform Kobo Toolbox, which was used to migrate the data collected with the ODK Collect application, also open-source. The resulting mosquito dataset was published as a Darwin Core Archive, a standardized format for sharing biodiversity data [26]. The core data table contains 3,840 records and includes an extension data table that provides additional information about the core records [27]. The dataset was submitted and validated using the Integrated Publishing Toolkit (IPT) validator tool available from the Global Biodiversity Information Facility (GBIF) [28]. Metadata fields are also available from the GBIF website [29].

## RESULTS

During the study period, 3,840 entomological occurrences (available in GBIF) were obtained from 960 records over 24 weeks at the 40 sentinel sites established in Praia, Cabo Verde. These occurrences included adult male and female mosquitoes of the species *Ae. aegypti* and *Cx. pipiens* s.l. collected with BG Sentinel traps, as well as *Ae. aegypti* eggs collected with OVTs.

The BG Sentinel adult trap collected 4,628 mosquitoes during the 24-week study period. *Cx. pipiens* s.l. was the most abundant species. *Ae. aegypti,* the vector of the Zika and dengue epidemics in Cabo Verde, accounted for 29.32% of the mosquitoes collected (Table 3). During the sampling, only seven BG Sentinel traps out of the 480 placed during the entire study period were lost due to blackout or disconnection of the power supply by the residents.

The OVTs collected a total of 100,967 *Ae. aegypti* eggs during the 24-week study period. This result shows the importance of this tool, not only for monitoring *Ae. aegypti,* but also as a physical measure by eliminating a considerable number of eggs from the environment, even in situations of low *Ae. aegypti* circulation [30, 31] (*Ae. aegypti* columns in Table 3). The average OI value obtained for all OVTs (*n* = 40) during the study period was 87.35%. All OVTs recorded values above the threshold considered a risk for vector control strategies (>10%) [32]. Specifically, 25% and 100% were the lowest and highest OI values found, respectively; only two OVTs had average OI values lower than 60%, indicating a very high infestation of *Ae. aegypti* in Praia, Cabo Verde (Table 3). Finally, during sampling, 15 of the 960 OVTs placed throughout the study period were lost because they were lying around, drying out, or missing.

We analyzed the sampling data to identify temporal relationships in the abundance of mosquitoes captured during the study period and in the values of the entomological indices. Varied temporal profiles of the collections and OI and EDI indices were obtained (Figure 3). The Pearson analysis showed a moderate positive correlation between OI and egg number (*r* = 0.68 and *p*-value = 0.0004). An increase in the values recorded for the number of eggs and OI was observed over time, with a more pronounced decrease for the former. The abundance of adult mosquitoes followed a cyclical profile, with two pronounced peaks in weeks 3–4 and 13–14. There was no relationship between the abundance of *Ae. aegypti* adults and eggs.

The abundance of mosquito vectors plays a key role in determining the timing and magnitude of arbovirus epidemics. Understanding the temporal relationship of mosquito abundance allows for the prediction of disease outbreaks, facilitating the implementation of vector control strategies [33].

**Table 3.** Mosquito abundance and mean entomological indices from entomological sampling for each sentinel OVT.

| OVT | No. adult mosquitoes | No. *Ae. aegypti* | No. *Cx. pipiens* s.l. | No. Eggs | OI% | EDI |
|---|---|---|---|---|---|---|
| OVT-1 | 143 | 31 | 112 | 999 | 86 | 49 |
| OVT-2 | 111 | 13 | 98 | 144 | 25 | 24 |
| OVT-3 | 79 | 41 | 38 | 1,772 | 91 | 84 |
| OVT-4 | 158 | 26 | 132 | 1,376 | 79 | 72 |
| OVT-5 | 71 | 36 | 35 | 2,641 | 95 | 125 |
| OVT-6 | 946 | 95 | 851 | 2,035 | 79 | 107 |
| OVT-7 | 148 | 29 | 119 | 790 | 54 | 60 |
| OVT-8 | 280 | 83 | 197 | 2,436 | 87 | 116 |
| OVT-9 | 8 | 3 | 5 | 3,958 | 83 | 198 |
| OVT-10 | 62 | 37 | 25 | 1,791 | 87 | 85 |
| OVT-11 | 7 | 3 | 4 | 796 | 70 | 46 |
| OVT-12 | 21 | 16 | 5 | 1,800 | 91 | 81 |
| OVT-13 | 56 | 22 | 34 | 2,697 | 83 | 134 |
| OVT-14 | 47 | 21 | 26 | 1,798 | 90 | 89 |
| OVT-15 | 63 | 38 | 25 | 3,299 | 91 | 149 |
| OVT-16 | 15 | 9 | 6 | 2,531 | 91 | 115 |
| OVT-17 | 27 | 16 | 11 | 1,801 | 87 | 85 |
| OVT-18 | 134 | 43 | 91 | 4,097 | 100 | 170 |
| OVT-19 | 148 | 36 | 112 | 2,106 | 87 | 100 |
| OVT-20 | 109 | 58 | 51 | 6,384 | 100 | 266 |
| OVT-21 | 223 | 33 | 190 | 2,243 | 100 | 93 |
| OVT-22 | 62 | 52 | 10 | 1,661 | 86 | 83 |
| OVT-23 | 25 | 15 | 10 | 993 | 65 | 66 |
| OVT-24 | 112 | 77 | 35 | 3,058 | 95 | 132 |
| OVT-25 | 25 | 11 | 14 | 2,656 | 91 | 120 |
| OVT-26 | 99 | 63 | 36 | 2,205 | 100 | 95 |
| OVT-27 | 73 | 44 | 29 | 3,327 | 100 | 138 |
| OVT-28 | 56 | 12 | 44 | 1,691 | 91 | 80 |
| OVT-29 | 168 | 70 | 98 | 4,278 | 100 | 178 |
| OVT-30 | 41 | 29 | 12 | 4,183 | 100 | 190 |
| OVT-31 | 221 | 52 | 169 | 1,716 | 100 | 71 |
| OVT-32 | 76 | 30 | 46 | 5,978 | 95 | 271 |
| OVT-33 | 144 | 41 | 103 | 3,367 | 100 | 140 |
| OVT-34 | 336 | 33 | 303 | 3,050 | 100 | 127 |
| OVT-35 | 46 | 2 | 44 | 1,005 | 95 | 43 |
| OVT-36 | 12 | 4 | 8 | 934 | 72 | 58 |
| OVT-37 | 59 | 38 | 21 | 540 | 66 | 33 |
| OVT-38 | 17 | 12 | 5 | 6,329 | 100 | 263 |
| OVT-39 | 94 | 65 | 29 | 4,588 | 91 | 208 |
| OVT-40 | 106 | 18 | 88 | 1,864 | 91 | 85 |
| **All OVT\*** | **4,628** | **1,357** | **3,271** | **100,917** | **87.35** | **115.73** |

\* "All OVT" reports the total values for the first four columns and the average values for the last two columns.

However, using OVT data has limitations. The number of eggs deposited in an OVT does not necessarily represent the abundance of biting female mosquitoes [34]. The Egg density EDI (Table 3) in this study is not linked to adult mosquito density; other factors like climate drivers (temperature, rainfall, and humidity) can affect the mosquito's life cycle and breeding site productivity [35–37].

## RE-USE POTENTIAL

The records and results presented in this study provide important information for future field surveillance studies, integrating indicators beyond entomological ones and providing

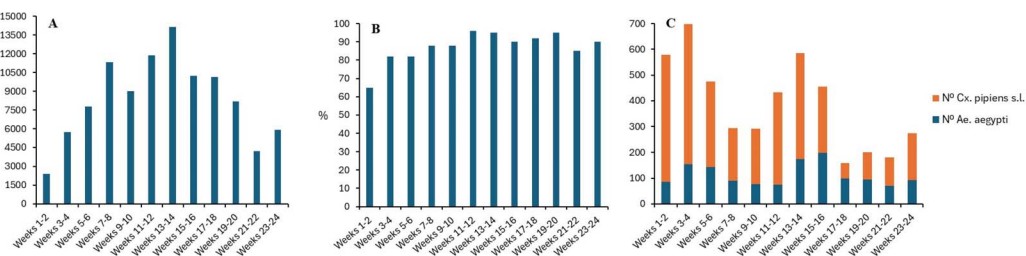

**Figure 3.** (A) Average number of eggs over time, (B) Average OI over time, (C) Average number of *Ae. aegypti* and *Cx. pipiens* s.l. mosquitoes over time.

a large and diverse number of records that can be analyzed, linked to this study, or compared with other areas in Cabo Verde or abroad.

Hence, collecting climatic, environmental, and sociodemographic datasets, as well as entomological data, with the potential to be analyzed in future studies, provides evidence for the creation of robust predictive surveillance systems that facilitate public health decision-making.

## DATA AVAILABILITY

The data supporting this article are published through the IPT of the Jean Piaget University of Cabo Verde, and are available, under a CCO waiver, in the GBIF repository [27].

## EDITOR'S NOTE

This paper is part of a series of Data Release articles working with GBIF and supported by TDR, the Special Program for Research and Training in Tropical Diseases, hosted at the World Health Organization [38].

## ABBREVIATIONS

EDI, Egg Density Index; GBIF, Global Biodiversity Information Facility; IPT, Integrated Publishing Toolkit; OI, Ovitrap Index; OVT, Ovitrap.

## DECLARATIONS

### Ethical approval

Not applicable.

### Consent publication

Not applicable.

### Competing interests

The authors declare that they have no competing interests.

### Authors' contributions

LFG: conceptualization, funding acquisition, project administration, data curation, writing (original draft, editing), data analysis, investigation, supervision, field data; KS: investigation, supervision, field data; BP: investigation, field data; PS: investigation,



resources (software); UC and JM: validation; HRR: investigation, supervision, field data, writing (review and editing). All authors read, revised, and approved the final manuscript.

## Funding

This work was partially supported by the 2nd edition of funding for research projects from the Cabinet of Higher Education, Science and Technology (GESCT) of the Ministry of Education of Cab Verde (Notice 0009).

## Acknowledgements

The authors would like to thank everybody who contributed to the creation of these datasets and paper, including all members of the Tropical Diseases Research Group at Jean Piaget University, for their continued support in the fieldwork and identification of entomological samples, without whom this study would not have been possible (a special mention to Deinilson Mendes, in memoriam); to Jean Piaget University for its support with transportation to the field; and to the other collaborating institutions that supported the first planning phase of the project: the National Institute of Territorial Management (INGT), the National Institute of Statistics (INE), and the University of Cabo Verde (UniCV). We would like to thank Paloma Helena Fernandes Shimabukuro and Tsiky Rabetrano for their help with data submission and facilitation in using the GBIF platform.

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
