## [Editor Report]

Editor’s AssessmentCabo Verde (or Cape Verde), is an island Nation off the coast of West Africa, and lies at the crossroads of three continents. Due to this location, climate change, and the effects of intense human and commercial trafficking, it is extremely vulnerable to the emergence and resurgence of vector-borne diseases. However systematic surveillance data on these vector species remains limited, hindering for entomological and modelling research and control strategies. This paper is one of a series of Data Release papers in GigaByte supported by TDR and the WHO describing datasets hosted in GBIF to tackle these data gaps in vectors of human disease data. This paper present the collection of entomological, demographic, water and sanitation, and climatic data in Praia, the capital of Cabo Verde and a hotspot for mosquito-borne diseases in the country. Data collected over 24 weeks from ovitraps at forty sentinel points and presenting 3,840 records from 4,628 mosquitoes. The records and results presented in this study provide important information for future field surveillance studies. Bringing together other indicators to providing a large and diverse number of records that can be analyzed, linked to this study, or compared with other areas in Cabo Verde or abroad.Editor’s AssessmentCabo Verde (or Cape Verde), is an island Nation off the coast of West Africa, and lies at the crossroads of three continents. Due to this location, climate change, and the effects of intense human and commercial trafficking, it is extremely vulnerable to the emergence and resurgence of vector-borne diseases. However systematic surveillance data on these vector species remains limited, hindering for entomological and modelling research and control strategies. This paper is one of a series of Data Release papers in GigaByte supported by TDR and the WHO describing datasets hosted in GBIF to tackle these data gaps in vectors of human disease data. This paper present the collection of entomological, demographic, water and sanitation, and climatic data in Praia, the capital of Cabo Verde and a hotspot for mosquito-borne diseases in the country. Data collected over 24 weeks from ovitraps at forty sentinel points and presenting 3,840 records from 4,628 mosquitoes. The records and results presented in this study provide important information for future field surveillance studies. Bringing together other indicators to providing a large and diverse number of records that can be analyzed, linked to this study, or compared with other areas in Cabo Verde or abroad.

---

## [Reviewer Report]

Reviewer name and names of any other individual's who aided in reviewer Yannan FanDo you understand and agree to our policy of having open and named reviews, and having your review included with the published papers. (If no, please inform the editor that you cannot review this manuscript.)YesIs the language of sufficient quality?YesPlease add additional comments on language quality to clarify if needed
Are all data available and do they match the descriptions in the paper? YesAdditional CommentsAre the data and metadata consistent with relevant minimum information or reporting standards? See GigaDB checklists for examples <a href="http://gigadb.org/site/guide" target="_blank">http://gigadb.org/site/guide</a>YesAdditional CommentsIs the data acquisition clear, complete and methodologically sound?YesAdditional CommentsIs there sufficient detail in the methods and data-processing steps to allow reproduction?YesAdditional CommentsIs there sufficient data validation and statistical analyses of data quality? YesAdditional CommentsIs the validation suitable for this type of data?YesAdditional CommentsIs there sufficient information for others to reuse this dataset or integrate it with other data?YesAdditional CommentsAny Additional Overall Comments to the AuthorRecommendationAccept

---

## [Reviewer Report]

Upload additional filesDRR-202509-03-R01/stage_files/DRR-202509-03/Review MS/gx-DR-1758071937(1).pdfReviewer name and names of any other individual's who aided in reviewer N.D.A.D.WijegunawardanaDo you understand and agree to our policy of having open and named reviews, and having your review included with the published papers. (If no, please inform the editor that you cannot review this manuscript.)YesIs the language of sufficient quality?YesPlease add additional comments on language quality to clarify if needed
Slight revisions are needed to improve the clarity of some long sentences.Are all data available and do they match the descriptions in the paper? YesAdditional CommentsIf possible, discuss further how understanding the temporal relationship in mosquito abundance is critical for predicting outbreak risks and optimizing vector control strategies.Are the data and metadata consistent with relevant minimum information or reporting standards? See GigaDB checklists for examples <a href="http://gigadb.org/site/guide" target="_blank">http://gigadb.org/site/guide</a>YesAdditional CommentsIs the data acquisition clear, complete and methodologically sound?YesAdditional CommentsIf possible, provide more details on how the 40 sentinel points were selected for sample collection, including the selection criteria.Is there sufficient detail in the methods and data-processing steps to allow reproduction?YesAdditional CommentsIs there sufficient data validation and statistical analyses of data quality? YesAdditional CommentsIs the validation suitable for this type of data?YesAdditional CommentsIs there sufficient information for others to reuse this dataset or integrate it with other data?YesAdditional CommentsAny Additional Overall Comments to the AuthorPlease review the article and correct any grammar and spelling mistakes, even if they seem minor.RecommendationMinor Revision